# Exploration of Challenges and Opportunities for Good Pharmacy Practices in Bangladesh: A Qualitative Study

**DOI:** 10.3390/pharmacy13010026

**Published:** 2025-02-13

**Authors:** Nantu Chakma, Sunjida Binta Ali, Md. Saimul Islam, Tanisha Momtaz, Noshin Farzana, Raian Amzad, Sharful Islam Khan, Md. Iftakhar Hassan Khan, Abul Kalam Azad, Zaheer-Ud-Din Babar, Aliya Naheed

**Affiliations:** 1Nutrition Research Division, International Centre for Diarrhoeal Disease Research, Bangladesh (icddr,b), Dhaka 1212, Bangladesh; nantu@icddrb.org (N.C.); saimul.islam@icddrb.org (M.S.I.); noshin.farzana@icddrb.org (N.F.); 2Health Systems and Population Studies Division, International Centre for Diarrhoeal Disease Research, Bangladesh (icddr,b), Dhaka 1212, Bangladesh; sunjida27@gmail.com (S.B.A.); istt87@gmail.com (T.M.); sharful@icddrb.org (S.I.K.); 3School of Pharmacy, BRAC University, Dhaka 1212, Bangladesh; 4Management Sciences for Health (MSH), Dhaka 1212, Bangladesh; amzad.raian@gmail.com (R.A.); ikhan.311266@gmail.com (M.I.H.K.); profakazad@gmail.com (A.K.A.); 5College of Pharmacy, Qatar University, Doha 2713, Qatar; z.babar@qu.edu.qa

**Keywords:** good pharmacy practice, model pharmacy, pharmacists, challenges, opportunities, Bangladesh

## Abstract

Background: In 2015, the Directorate General of Drug Administration (DGDA) of Bangladesh accredited model pharmacies (MPs) to enhance the quality of pharmacy services across the country. We examined the challenges and opportunities for pharmacists in MPs, and also explored the perspectives of the pharmacy stakeholders for improving good pharmacy practices (GPPs) in Bangladesh. Methods: In-depth interviews (IDIs) were conducted with graduate pharmacists (Grade A) and diploma pharmacists (Grade B) recruited from a few selected MPs that were included in a previous study. Key informant interviews (KIIs) were conducted with the government and non-government stakeholders who were involved in pharmacy regulations and practices. Trained qualitative researchers conducted IDIs and KIIs using interview topic guides under relevant themes developed by the study investigators. Results: Between February and March 2021, nine Grade A and six Grade B pharmacists and nine government and non-government stakeholders were interviewed. The key challenges, as well as demotivational factors, for Grade A pharmacists were reported to be multiple responsibilities, inadequate salary, poor social status, an unfavorable working environment, long working hours, a lack of recognition, and low respect for their profession. However, Grade B pharmacists expressed job satisfaction, primarily due to working opportunities in reputable pharmacies and learning opportunities. The stakeholders reported a high operation cost of the MPs, a shortage of trained pharmacists, poor salary structures, and a lack of public awareness about the critical roles of the pharmacists in healthcare to be challenges of retaining Grade A pharmacists at the MPs. Addressing the challenges of the pharmacists and revising compensation packages along with strengthening monitoring systems would be important for improving GPPs at the MPs. Conclusions: This study has demonstrated that specifying the roles of the pharmacists, offering competitive packages, conducive working hours, and professional recognition would be imperative for the retention of trained pharmacists at MPs. Implementing regulatory standards and monitoring performance would enhance good pharmacy practices in Bangladesh.

## 1. Introduction

The World Health Organization (WHO) has estimated that there are over 2.1 million pharmacists and other pharmaceutical professionals in the world [1]. Before the mid-1990s, the role of pharmacists was restricted to labeling, compounding, and dispensing medicine [2]. Currently, pharmacists play a vital role in healthcare, adopting a patient-centered approach [3], and they are often the first point of contact of a patient with the healthcare system [4]. Pharmacists have to perform several roles, including filling prescriptions, handling orders, checking inventory, maintaining patients’ records, counseling patients, etc., and these roles vary from country to country based on the levels of competencies [5,6]. Pharmacists play a very important role in ensuring the proper distribution and use of medicines [7]. It is imperative that pharmacists follow good pharmacy practices (GPPs) in their professional lives, an international standard for all pharmacists set by the WHO [3].

However, pharmacists often face ethical, economic, and legal issues in their day-to-day work, which create a gap between what is expected from them and what is being done. There are concerns about the quality of pharmacy practices, particularly in low- and middle-income countries (LMICs) due to poor regulations and policies [8], albeit limited information is available about the challenges of pharmacists in pharmacy practices in the LMICs. A scarcity of graduate pharmacists in retail pharmacies has been identified in Yemen, coupled with dissatisfaction with the job, poor salaries, and a lack of regulations and standards [9]. A poor working environment and inadequate training and knowledge were reported to be challenges in pharmacy practices in Indonesia [10].

In Bangladesh, it has been estimated that there are around 191,512 registered retail pharmacies [11], and an equal number of pharmacies operate without registration [12]. There are three categories of pharmacists in Bangladesh: (i) the Grade A pharmacists, who have a graduate degree in pharmacy offered by the recognized higher degree institutions, typically universities; (ii) the Grade B pharmacists, who have completed a 3-year Diploma in Pharmacy course offered by the Pharmacy Council of Bangladesh (PCB) under the recognized paramedic institutions; and (iii) the Grade C pharmacists, who have completed a 3-month training course on dispensing medicine jointly offered by the PCB and the Bangladesh Chemist and Druggist Samity (BCDS), an organization of drug dispensers in Bangladesh [11]. The PCB is the governing body in Bangladesh for ensuring quality pharmacy education, capacity building, and registration of pharmacists. The Directorate General of Drug Administration (DGDA) is the government regulatory body for ensuring quality pharmacy practices in Bangladesh. In Bangladesh, independent prescribing rights for pharmacists are limited, except for over-the-counter (OTC) drugs. There are 39 OTC drugs in Bangladesh [13]. As per the law of Bangladesh, any individual having a valid pharmacist registration of Grade A, Grade B, or Grade C can apply for a license from the DGDA to open a retail pharmacy and sell OTC and prescription medicines. These regular pharmacies often operate with less oversight, lack graduate pharmacists, sometimes sell prescription medicines without prescriptions, have an absence of counseling services, and are often run by Grade C pharmacists [14,15,16].

In 2015, the DGDA launched accreditation programs for two new levels of medicine outlets, the model pharmacy and the model medicine shop, in order to promote good pharmacy practices (GPPs). A model pharmacy (MP) is managed, served, and supervised by a graduate pharmacist (Grade A) with the support of lower-grade pharmacists, such as Grade B [17]. Between 2016 and 2020, the DGDA inaugurated 274 model pharmacies in Bangladesh to ensure high-quality pharmacy services, requiring the presence of graduate pharmacists and adherence to the strict guidelines for dispensing medicines (e.g., labeling), and the presence of storage, pharmacy-grade refrigerators, air conditioning systems, adequate space (at least 300 square feet), patient counseling services, etc. MPs are designated to promote safe pharmacy practices and prevent the misuse of antibiotics [17,18], whereas ordinary pharmacies are often operated with less oversight, with a lack of graduate pharmacists and an absence of counseling services, and are often run by Grade C pharmacists [12,14]. However, after 2020, many MPs have failed to retain trained pharmacists, which is an important barrier to promoting GPPs [19], albeit challenges faced by the pharmacists have not been explored in Bangladesh. We have explored potential challenges and opportunities for GPPs in the model pharmacies in Bangladesh from the perspectives of pharmacists and the relevant stakeholders.

## 2. Materials and Methods

### 2.1. Study Design and Settings

This study was conducted in five districts (Dhaka, Chattogram, Rangpur, Barisal, Khulna), where 56% of MPs were inaugurated by the DGDA. A list of Grade A and Grade B pharmacists was developed by visiting the model pharmacies that were randomly selected from the list of MPs inaugurated by the DGDA. A pharmacist was recruited from the selected MPs if they met the following criteria: (i) have a bachelor’s degree in Pharmacy (Grade A) or a Diploma in Pharmacy degree (Grade B), (ii) have a valid registration either as a Grade A or a Grade B pharmacist from the PCB, and (iii) have at least 6 months of working experience in an MP. We purposefully selected from the list of eligible Grade A and Grade B pharmacists to have a good mix of age, sex, location, and duration of employment before inviting them to participate in an in-depth interview (IDI). Simultaneously, an additional list of the government and non-government stakeholders was created in consultation with the DGDA for conducting key informant interviews (KIIs), including those stakeholders who have been involved in regulatory or administrative or human resource development roles for pharmacy practice in Bangladesh, such as representatives of the DGDA, the PCB, institutions offering degrees or a Diploma in Pharmacy, and other relevant organizations.

### 2.2. Data Collection

IDIs with Grade A and Grade B pharmacists were conducted face-to-face at their duty stations by trained team members following an IDI topic guide (Appendix A) that included the participants’ general information, attitudes, opinions about the model pharmacy initiative, responsibilities, job satisfaction or dissatisfaction, motivational or demotivational factors, challenges, and opportunities for continuing services in an MP. The IDI topic guide also explored recommendations of the pharmacists for the retention of trained pharmacists in MPs in the long term. A separate interview topic guide (Appendix A) was used for conducting KIIs with the stakeholders to explore their perspectives about the challenges of the model pharmacy initiative, opportunities, and recommendations for retaining Grade A pharmacists in the MPs. Both the IDIs and KIIs were audio recorded following consent.

### 2.3. Data Analysis

The audio records of each IDI and KII were transcribed line by line in the local language (Bangla) in order to generate a transcript, and the transcript was coded using a code list developed around the interview topic guides by a bilingual researcher (NC) guided by the Principal Investigator (AN). Atlas.ti (version 7.5) software was used for coding. Codes and sub-codes were used for data extraction. Extracted data were first summarized in Bangla following a thematic approach based on key themes and sub-themes before being translated into English. Any quote of a participant was labeled by the type of interview (IDI/KII), type of the respondent (labeled as ‘A’ for a Grade A pharmacist, and ‘B’ for a Grade B pharmacist), and district name (BA for Barishal, CH for Chattogram, DH for Dhaka, RA for Rangpur, and KH for Khulna). A unique interview number was assigned to each participant for de-identification and maintaining data privacy.

The transcripts were analyzed to present the summary results under two broad themes: ‘theme i’, challenges of model pharmacies, and ‘theme ii’, opportunities to improve model pharmacies. A few sub-themes were generated as guided by the analyses of the transcripts. The sub-themes under ‘theme I’ included multiple responsibilities of pharmacists, job dissatisfaction of the pharmacists, demotivational factors, operational costs of model pharmacies, shortage of qualified pharmacists, and low social status. The sub-themes under ‘theme ii’ included salary and benefits, the role of the pharmacist-in -charge, and monitoring of the MPs. The summary results of the sub-themes of the IDIs were collated to explore the views of the pharmacists about challenges and opportunities extrapolated from their own experiences to draw recommendations for their retention in the MPs. The summary results of the sub-themes of the KIIs were collated to explore the views of the stakeholders about the challenges and opportunities of the pharmacists for improvement.

### 2.4. Ethics Approval

Ethics approval was obtained from the Ethical Review Committee (ERC) of the International Centre for Diarrhoeal Disease Research, Bangladesh (icddr,b PR-20142, 15 February 2021) and the National Research Ethics Committee of the Bangladesh Medical Research Council (BMRC). Written informed consent was obtained from each respondent prior to enrolment in the study and before conducting an interview.

## 3. Results

Between February and March 2021, IDIs were conducted with nine Grade A pharmacists and six Grade B pharmacists, and Table 1 summarizes their socio-demographic characteristics. The age range of the participants was 20–29 years, and they were purposively selected from five districts where 56% of the MPs were established. The majority of Grade A pharmacists were female, and the majority of Grade B pharmacists were male. The length of the jobs of the pharmacists working at a model pharmacy ranged from six months to three years.

KIIs were conducted with nine government and non-government stakeholders. The majority of the stakeholders were male, with an age range of 31–61 years. The key informants were mostly from Dhaka, where the majority of the pharmacy regulatory agencies were located, and a small number of key informants were recruited from the Khulna district (Table 2).

### 3.1. Challenges of Model Pharmacies: Perspectives of the Pharmacists

#### 3.1.1. Multiple Responsibilities

Grade A pharmacists reported a wide range of responsibilities in the model pharmacies, including reviewing prescriptions and patient counseling to ensure proper medicine use, and verifying medicines dispensed by the pharmacy staff to ensure alignment with the doctor’s prescriptions, with particular attention to antibiotics. Grade A pharmacists also ensured the proper storage of medicines at a proper location, maintenance of the right temperature, and regular monitoring of both room and refrigerator conditions. Another key responsibility was to supervise the junior staff dispensing medicines to the customers and oversee overall pharmacy operations. Additionally, Grade A pharmacists were responsible for tracking the expiration dates of stock medicines and returning expired medicines to the medicine companies, while procuring new stock. A few Grade A pharmacists reported that they manage the cash counter and measure blood pressure and blood sugar levels in addition to their regular responsibilities in the MP. One Grade A pharmacist stated the following:

“If needed, I have to help them (other staff at the pharmacy) and observe what they are doing. Sometimes, I also have to manage cash when necessary.”(IDI-A-DH-05)

The Grade B pharmacists reported similar responsibilities to that of a Grade A pharmacist, except that some reported that they occasionally performed data entry tasks for which they were not responsible. One Grade B pharmacist stated the following:

“……on top of my other works, I also sometimes perform data entry activities at the pharmacy”.(IDI-B-CH-06)

#### 3.1.2. Job Dissatisfaction

The majority of the Grade A pharmacists expressed dissatisfaction with their jobs in the model pharmacies. The key reasons for dissatisfaction were low salaries, negative public perceptions about the pharmacy profession, and a lack of recognition. Additionally, it was perceived by the Grade A pharmacists that they do not get the respect they deserve from the owners of the model pharmacies, which aggravated their dissatisfaction further. Long working hours and no compensation for overtime were reported to be significant reasons for dissatisfaction. Furthermore, owners occasionally insisted on prioritizing selling medicines over providing proper counseling to customers, and they have limited autonomy to make choices that may differ from those of the pharmacy owners.

“The salary structure is very poor. When I joined here (model pharmacy) initially, they offered me ten thousand takas (equivalent to USD 85) per month. They said this was because I would have fewer duties. I thought this amount was too low, but later I decided that I needed the experience, so I joined. They (model pharmacy authority) said they would increase my salary after three months but they didn’t. They didn’t keep their word and paid only the initial salaries.”(IDI-A-DH-05)

The majority of the Grade B pharmacists reported being satisfied with their job in the model pharmacies. Factors contributing to their satisfaction included working in a reputable pharmacy, maintaining good relationships with colleagues, having opportunities to learn, and improving their skills in dispensing medicines.

“I am satisfied, because considering the current job market, this is a good job. I do my duty for eight hours and I work in a good institution, which is one of the good companies in Bangladesh”.(IDI-B-RA-01)

A few Grade B pharmacists expressed dissatisfaction due to low salaries and insufficient benefits. They also felt that sometimes they had a lack of knowledge about basic pharmaceutical issues, such as a lack of understanding of the generic names of the medicines, which contributed to dissatisfaction for some Grade B pharmacists.

#### 3.1.3. Demotivational Factors

Several factors contributed to the demotivation of Grade A pharmacists to work in a model pharmacy. First, they often felt undervalued in the workplace, and some believed that they did not receive the respect or the recognition of their position from their co-workers or customers or the pharmacy owners that they deserved as holders of a graduate degree in pharmacy. Second, the work environment was described as less conducive in terms of inadequate salaries and an absence of any benefits package, which highly demotivated the Grade A pharmacists from continuing their roles in MPs. Third, job insecurity and limited job opportunities led to demotivation significantly. Fourth, Grade A pharmacists were often recruited as medicine sellers, which did not fall under their professional responsibilities, and this demotivated them to continue working in a model pharmacy. One Grade A pharmacist stated the following:

“Some of my colleagues treat me as a salesman, and they don’t want to understand the role of a pharmacist. They even don’t know that a Grade A pharmacist is essential to run a model pharmacy.”(IDI-A-DH-05)

A few Grade A pharmacists reported that they were compelled to work on government holidays, which disrupted their work–life balance and left them with little time to spend with their families, which negatively impacted their personal lives. The constant demand for unregulated long working hours contributed to feelings of burnout and dissatisfaction with their jobs at the model pharmacies. In addition, Grade A pharmacists generally believed that very few people truly understood the professional roles of a Grade A pharmacist, and, as a result, the general public often did not show respect toward pharmacists working in medicine outlets. This lack of recognition led to significant challenges in their social lives and even impacted their personal relationships. Some Grade A pharmacists also expressed that they did not consider their pharmacy job to be prestigious enough to meet their career aspirations, which was a demotivational factor.

“Social status is an issue. For example, if I work for a company (pharmaceutical) I may be treated well as a pharmacist but now I am a shopkeeper. Obviously, a shopkeeper, as a result, I become underrated.”(IDI-A-BA-08)

Grade B pharmacists similarly expressed dissatisfaction with their poor salary, inadequate benefits, and low social status as demotivating them to work in a model pharmacy. Further, despite being frontline healthcare providers, there is no provision for health insurance coverage for the pharmacists, which is a driver for demotivation.

“The bad feeling is, our profession (pharmacist) is closely connected with the health service and we have to work in the frontline but we didn’t see any initiative from the government. We don’t have any health insurance facility and there is no trust in this profession.”(IDI-B-DH-03)

### 3.2. Challenges of Model Pharmacies: Perspectives of Stakeholders

#### 3.2.1. Operation Cost

Stakeholders have identified several challenges associated with operating model pharmacies in Bangladesh. First, the cost of running a model pharmacy is significantly higher, while the profit margins are relatively low. Second, hiring a Grade A pharmacist is expensive for the pharmacy owners. In addition to hiring a Grade A pharmacist, owners are also required to hire drug sellers, which further increases the operational costs. As a result, it becomes financially difficult for owners to afford a Grade A pharmacist to run an MP.

“If you want to do all the package practice in an appropriate way, it comes at a cost. The costs of sustaining a Grade A pharmacist are prohibitively high if all the rules of a model pharmacy are to be followed.”(KII-DHA-06)

#### 3.2.2. Shortage of Qualified Pharmacists

Most of the stakeholders highlighted the shortage of qualified pharmacists in Bangladesh as a major challenge for operating the model pharmacies. Although the demand for quality pharmacy services is high due to a growing population and increasing healthcare needs, the production of qualified pharmacists is insufficient to meet this demand. As a result, most Grade A pharmacists prefer to work in the pharmaceutical industry, where they get better benefits packages and other opportunities, as opposed to working in retail pharmacies. One stakeholder stated the following:

“The head of a model pharmacy is Grade A pharmacist and assisted by a Grade B pharmacist. These two types of pharmacists are not adequately produced in our country.”(KII-DH-03)

#### 3.2.3. Low Social Status

Stakeholders reported that the low social status of the Grade A pharmacists is a significant concern. When graduate pharmacists join a pharmacy, they are often expected to focus on selling medicines rather than utilizing their professional skills. This misalignment between their training and the actual duties they perform leads to a sense of underappreciation, among both the customers and their own professional communities. This lack of respect and recognition contributes to their dissatisfaction with the role. One of the stakeholders stated the following:

“The society can think that the (pharmacists) is only working in a pharmacy. A Grade A pharmacist had to take a graduate degree at least. In our country, we have only a few pharmacies where pharmacists have a place for counseling.”(KII-KH-02)

Some of the stakeholders also cited that a poor salary structure for the Grade A pharmacists is a major reason for the discontinuation of jobs in MPs. A few stakeholders argued that there is a high turnover of Grade A pharmacists in the model pharmacies because the Grade A pharmacists are lured by the pharmaceutical industry with an offering of better benefits and career opportunities.

### 3.3. Opportunities to Improve Model Pharmacies: Perspectives of the Pharmacists

#### 3.3.1. Salary and Benefits

There are opportunities to improve the retention of Grade A pharmacists in the model pharmacies by offering competitive salaries and benefits. Most of the Grade A pharmacists suggested that the government could encourage model pharmacy owners to implement a competitive salary structure for the Grade A pharmacists backed by compensation packages and additional benefits, such as transportation and housing allowances, which would, in return, also encourage the model pharmacy owners to retain skilled pharmacists. One Grade A pharmacist stated the following:

“…Everybody wants to see self-benefit. If there is lesser salary provided in a model pharmacy than a pharmaceutical company why pharmacists would be interested to join in a model pharmacy?”(IDI-A-RA-04)

Grade B pharmacists similarly emphasized the importance of offering competitive salaries to Grade A and Grade B pharmacists to ensure their long-term retention and sustainability in the profession. They also proposed adding more benefits, including annual salary increments, fixed working hours, festival bonuses, and guaranteed weekends, as well as annual leaves to improve the chances of retention of Grade A pharmacists in MPs.

“We need a government policy for the pharmacists. Since the government has initiated a model pharmacy and model medicine shop program, there is an opportunity for Grade A and Grade B pharmacists. Therefore, there needs to be a policy for the salary and other facilities for staff. If this happens, pharmacists would show their interests.”(IDI-B-CH-06)

#### 3.3.2. Role of Pharmacist-in-Charge

Given the shortage of Grade A pharmacists, it has been explored whether Grade B pharmacists could fill in for Grade A pharmacists when necessary. However, the majority of the Grade A pharmacists indicated that Grade B pharmacists study a three-year diploma course, whose curriculum has comparatively low standards compared to the four-year bachelor’s course studied by a Grade A pharmacist. As a result, there is insufficient coverage of pharmaceutical topics in the course curriculum for the Grade B pharmacists. Examples include managing the extensive inventory of medicines and overseeing pharmacy operations, which would require a comprehensive knowledge of pharmacology, which Grade B pharmacists cannot learn from their limited diploma courses. Further, the current operational guidelines for the model pharmacies do not permit Grade B pharmacists to take on the role of a pharmacist-in-charge. As such, Grade B pharmacists are not equipped with adequate knowledge and training to assume the role of a pharmacist-in-charge in a model pharmacy, which is the expected role of a Grade A pharmacist. Therefore, the Grade A pharmacists discouraged the promotion of the role of a Grade B pharmacist to the role of a Grade A pharmacist, and instead advised that the number of graduate pharmacists be increased to facilitate the operations of a high number of model pharmacies in order to improve quality of pharmacy practices in Bangladesh.

“There is no opportunity to replace Grade A pharmacist by Grade B. I guess, there is no opportunity to run a model pharmacy without a Grade A pharmacist in-charge. Otherwise, those pharmacies will not be considered as model pharmacies. One Grade A pharmacist is essential to run a model pharmacy.”(IDI-A-DH-07)

Nonetheless, a few Grade A pharmacists acknowledged that the Grade B pharmacists could potentially assume some limited roles of a pharmacist-in-charge in the absence of a Grade A pharmacist, provided that they acquire proper training and gain adequate experience. On the contrary, the Grade B pharmacists expressed confidence in their ability to take on the role of a pharmacist-in-charge, and the majority of them argued that their qualifications, including the length of their studies and the hands-on training received during an internship, adequately prepare them for taking over the role of a Grade A pharmacist. One Grade B pharmacist stated the following:

“If you ask about the options of Grade A pharmacists, I should say yes. Grade B pharmacists can manage model pharmacy because they have a 3 years 6 months duration pharmacy course and gain experience by doing an internship in a hospital setting. I don’t see any difference regarding the responsibility in the model pharmacy. They can manage various crisis moments and they can provide simple treatments, so I think Grade B pharmacists are capable to assume the role of a Grade A. Grade A pharmacist responsibilities can be easily performed by the Grade B pharmacists.”(IDI-B-DH-04)

### 3.4. Opportunities to Improve Model Pharmacy Services: Perspectives of the Stakeholders

#### 3.4.1. Salary and Benefits

The stakeholders recommended competitive salary structures, ensuring proper recognition and respect for pharmacists, and providing a congenial work environment for the pharmacists in order to foster accountability in pharmacy practices. Some stakeholders highlighted that offering competitive salaries would not only retain pharmacists but also elevate their social status.

“I believe to retain Grade A pharmacists, their salaries and benefits should be increased. At the same time, the government needs to take initiatives to raise awareness on the importance of qualified pharmacists in managing model pharmacies. Enhancing their economic recognition will also contribute to improving their social status.” (KII-DH-06)

#### 3.4.2. Role of Pharmacist-in-Charge

The stakeholders expressed mixed opinions about whether Grade B pharmacists should be allowed to assume the role of pharmacist-in-charge in model pharmacies. Some stakeholders opposed the idea by referring to the limited knowledge of Grade B pharmacists and their scarcity in numbers, which they believe would hinder their ability to meet the demanding task of managing a model pharmacy. However, a few stakeholders recognized the challenges in retaining Grade A pharmacists in this role and suggested improving the Grade B course curriculum to reduce the knowledge and skills gap of the Grade B pharmacists, so that they can take on the role of a pharmacist-in-charge when necessary. However, Grade A pharmacists receive proper training on professional ethics and adhere to them, which is a strong justification for recommending Grade A pharmacists to hold the role of a pharmacist-in-charge in a model pharmacy.

“The work of Grade A pharmacist cannot be performed by the Grade B pharmacist. Meanwhile, the number of Grade B pharmacists enrolling per year is less, compared to the number of Grade A pharmacists. Actually, there is no chance of substituting Grade A pharmacists. There are a few things that need to be done. One is that they have professional ethics, they have to have a place to practice ethics. Grade A pharmacists do not continue a job in a model pharmacy, because people are not being able to do any unethical practices with them. That’s why business is losing, that’s why they are not recruiting Grade A pharmacists.”(KII- DH-01)

#### 3.4.3. Monitoring of the Model Pharmacies

A few stakeholders recommended conducting regular monitoring of model pharmacies to ensure the presence of qualified pharmacists. This is because some owners do not hire Grade A pharmacists even after securing a model pharmacy license, which represents non-adherence to the model pharmacy standards. Robust monitoring and administrative oversight are recommended by the stakeholders to enforce standardized practices in model pharmacies. Such measures would compel owners to comply with the established guidelines and contribute to sustaining quality pharmacy services.

“The DGDA has to be strict. They have to check whether the Grade A pharmacists are available at the model pharmacies or not. Whether they are doing their jobs or not. It has been seen that many model pharmacies do not replace the vacancy of a pharmacists with a Grade A pharmacist after departure of the originally recruited Grade A pharmacist.”(KII-DH-09)

Additionally, stakeholders suggested maintaining quality standards and offering government support, such as, loan facilities for model pharmacy owners. This financial assistance would enable owners to recruit qualified Grade A pharmacists and strengthen the overall workforce quality in model pharmacies.

## 4. Discussion

The purpose of this study was to explore the challenges and opportunities for good pharmacy practices in model pharmacies in Bangladesh from the perspectives of pharmacists and other relevant stakeholders. Grade A pharmacists face several challenges that hinder their motivation and job satisfaction. These include juggling multiple responsibilities, receiving inadequate salaries, having a poor social status, working in unfavorable environments with long hours, and experiencing a lack of recognition or respect for their profession. In contrast, Grade B pharmacists reported job satisfaction, primarily due to working opportunities in reputable pharmacies and learning opportunities.

The stakeholders identified the high operational costs, the shortage of trained pharmacists, poor salary structures, and limited public awareness of pharmacists’ critical roles in healthcare as the key barriers to retaining Grade A pharmacists at model pharmacies. Stakeholders further recommended revising compensation packages, promoting qualified pharmacists, and strengthening monitoring systems to uphold and enhance the standards of pharmacy practice.

The study revealed that multiple responsibilities tend to impose a burden of workload on Grade A and Grade B pharmacists in model pharmacies, which poses a significant challenge that is similar to that of pharmacists in low- and middle-income countries (LMICs) [20,21,22]. Job dissatisfaction was highly related to the motivation levels of Grade A and Grade B pharmacists. Those who reported job satisfaction were typically employed in reputable model pharmacies, where they received better salaries and benefits compared to those working in less reputable pharmacies. Salaries and other benefits were also found to be key motivational factors for pharmacists in other LMICs [23]. In contrast, negative social perceptions, poor working conditions, inadequate facilities, lack of respect, and the desire to pursue careers in the pharmaceutical industry were cited as reasons for leaving model pharmacies for Grade A pharmacists.

Stakeholders also highlighted low salaries, poor facilities, low social status, family demands, and unfavorable working conditions as key reasons for the demotivation and discontinuation of Grade A pharmacists. These findings align with previous studies, which identified attractive salaries, opportunities for promotion, and professional recognition as some of the most powerful motivational factors for pharmacists [24,25].

Low social status and professional discontent among Grade A pharmacists present another significant challenge. Limited literatures on healthcare workforce dynamics indicates that professional identity and job satisfaction are closely linked to social recognition and the appropriate utilization of skills [25]. A lack of recognition from society and customers, coupled with a limited scope for professional growth in retail pharmacies, has led many pharmacists to prefer working in the pharmaceutical industry, where the benefits, salary structure, and career prospects are more appealing.

According to the model pharmacy accreditation guidelines, model pharmacies are required to have Grade A pharmacists [17]. Due to the high turnover of Grade A pharmacists, the possibility of shifting Grade B pharmacists to assume the role of pharmacist-in-charge in model pharmacies was explored. There were differences observed regarding the opportunities for making Grade B pharmacists in charge in model pharmacies among Grade A pharmacists and stakeholders and Grade B pharmacists. Grade A pharmacists and stakeholders were opposed to the idea of placing Grade B pharmacists in charge in model pharmacies, considering the insufficient training and inadequate curriculum of Grade B pharmacists. However, Grade B pharmacists found themselves to be qualified for the position, as they are capable of managing the routine tasks carried out by Grade A pharmacists, such as temperature management, medicine procurement, patient counseling, and staff supervision.

Despite the challenges, the opportunities to improve model pharmacy services in Bangladesh are primarily centered around enhancing salary structures, promoting qualified pharmacists, and establishing a robust monitoring system. A key recommendation made by both Grade A and Grade B pharmacists is to increase salaries and benefits, as competitive compensation is critical for retaining skilled pharmacists. Studies on healthcare workforce retention highlight that job satisfaction is closely tied to both intrinsic factors like professional recognition and extrinsic factors like financial compensation [26]. Studies on regulatory oversight in healthcare have found that consistent monitoring is essential for enforcing compliance and maintaining high service standards [27]. The stakeholders in Bangladesh pointed to the need for a proper monitoring system to ensure compliance with model pharmacy standards and the maintenance of the quality and integrity of pharmacy services.

Both the pharmacists and the stakeholders considered the model pharmacy initiative as a promising approach that could significantly enhance pharmacy practices. This type of accreditation of medicine outlets was found to be scalable and sustainable in Tanzania, as it effectively improved the quality of pharmacy services, ensured better adherence to regulatory standards, enhanced customer trust, and contributed to improved healthcare outcomes [28,29].

## 5. Strength and Limitations

The first limitation of this study is that the number of MPs was much higher in the Dhaka district, and all key regulatory agencies were located there; hence, the majority of our interviews were conducted in Dhaka, which might have reduced our ability to obtain a much broader perspective on the pharmacists and stakeholders working outside Dhaka. Second, due to a low number of Grade B pharmacists in the professional group, we were unable to increase the number of interviews with Grade B pharmacists. However, this study included a wide mix of pharmacists and stakeholders, including representatives from regulatory bodies, academicians, pharmacy owners, and representatives from medicine sellers’ organizations, which rendered a better representation of the pharmacy community in Bangladesh. Further, despite the low number of interviews, we were able to reach data saturation with the IDIs and KIIs of each group. This allowed us to obtain a broader knowledge base on the practical challenges of attaining good pharmacy practices in the MPs and potential opportunities for strengthening model pharmacy initiatives in Bangladesh.

## 6. Conclusions

This study has highlighted the challenges faced by Grade A and Grade B pharmacists working in model pharmacies. The model pharmacy initiative is a unique and promising effort to promote good pharmacy practices across Bangladesh. However, its success depends on rigorous monitoring, consistent enforcement, and support from the relevant authorities to ensure adherence to established standards. Despite the challenges, the model pharmacy initiative has great potential for further improvements by enhancing regulatory standards and effective implementation and fostering a supportive work environment in MPs for retaining pharmacists. Periodic training, better incentives, and the provision of heightened social value of the pharmacists are essential for empowering the pharmacists to deliver high-quality pharmacy services and promote better health outcomes for the people of Bangladesh.

## Figures and Tables

**Table 1 pharmacy-13-00026-t001:** Characteristics of the pharmacists.

Characteristics		Grade A (*n* = 9)	Grade B (*n* =6)
Gender	Male	3	6
	Female	6	0
Age group	20–24 years	2	3
	25–29 years	7	3
Location	Dhaka	5	3
	Rangpur	2	2
	Barisal	1	0
	Khulna	1	0
	Chattogram	0	1
Dispensing experience	6 months–1 year	2	1
	1–2 years	3	3
	2–3 years	2	1
	3 years	2	1
Type of involvement	Employee	9	6

**Table 2 pharmacy-13-00026-t002:** Characteristics of the stakeholders.

Characteristics		Total Number = 9
Gender	Male	8
	Female	1
Age (years)	31–40 years	2
	41–50 years	2
	51–60 years	3
	61 years and above	2
Education	HSC	1
	Bachelor	2
	Master	5
	Doctorate	1
Institution	Directorate General of Drug Administration (DGDA)	3
	Pharmacy owner	1
	Pharmacy Council of Bangladesh (PCB)	1
	Institute of Health Technology (IHT), Mohakhali	1
	Dhaka University	1
	Bangladesh Chemist and Druggist Samity (BCDS)	2
Location	Dhaka	7
	Khulna	2

## Data Availability

Data are accessed on a reasonable request to the corresponding author.

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
