# Peer review of "Exploration of Challenges and Opportunities for Good Pharmacy Practices in Bangladesh: A Qualitative Study"

_pharmacy, 2025, doi:10.3390/pharmacy13010026_

Round 1
Reviewer 1 Report
Comments and Suggestions for Authors
Manuscript ID: pharmacy-3338719
This is an interesting qualitative study on the challenges and opportunities of good pharmacy practice in Bangladesh. The work methodology is clear. The techniques used for data collection and processing also meet the scientific criterion. However, there are a few areas where it could be improved:
What is particularly distracting when reading a manuscript are technical and grammatical errors that require attention (e.g., inadequate spacing between letters and lines, lowercase initial letters in sentences, inadequate text alignment in the Introduction, good pharmacy practice or Good Pharmacy Practice in the text of the manuscript, etc.).
The name (e.g. Pharmacy Council of Bangladesh) and appropriate acronym (e.g. PCB) must be written in full when first described in the text, then only the acronym is used (e.g. lines 67 and 70).
Material and Methods in Abstract section require more data to give readers a better insight into this qualitative study (e.g. number of included respondents, study duration timeframe, etc).
The terms “IDI” (In Depth Interview) and “KII” (Key Informant Interview) must be described in more detail when they are mentioned for the first time in the Material and Methods section.
In the Material and Methods section, it is necessary to describe the time period when the research was conducted.
Line 137-138. This sentence requires correction to make it more understandable.
The Ethics approval subsection in Materials and Methods is missing the numbers and dates under which decisions granting consent to conduct research were recorded.
The first few sentences in the Discussion section should contain the purpose of the research and the author's summary of the study that was done.
Line 414-415. Sentence "Additional unsatisfactory aspects such as- low salaries, poor social status, lack of recognition, and long working hours." it seems that it is not complete and finished.
The Conclusions section should follow the research results, which means it should be more detailed.
Author Response
Response to Reviewer 1 Comments
We appreciate the comments and feedback made by reviewer 1. We have responded to each comment raised by the reviewer 1 and addressed the specific feedback point by point. Our responses are structured as follows:
a) Comments from the reviewer are presented in bold and
b) Our responses are provided directly under each comment.
Comments and responses:
- This is an interesting qualitative study on the challenges and opportunities of good pharmacy practice in Bangladesh. The work methodology is clear. The techniques used for data collection and processing also meet the scientific criterion. However, there are a few areas where it could be improved:
What is particularly distracting when reading a manuscript are technical and grammatical errors that require attention (e.g., inadequate spacing between letters and lines, lowercase initial letters in sentences, inadequate text alignment in the Introduction, good pharmacy practice or Good Pharmacy Practice in the text of the manuscript, etc.).
Response: We apologize for the editorial errors that remained in the version submitted. We highly appreciate pointing out those errors. Now we have thoroughly revised the manuscript to correct the grammatical errors, spacing, spelling mistakes, text alignments, and uppercase/ lowercase issues throughout the manuscript.
- The name (e.g. Pharmacy Council of Bangladesh) and appropriate acronym (e.g. PCB) must be written in full when first described in the text, then only the acronym is used (e.g. lines 67 and 70).
Response: We fully agree with this feedback. We have now elaborated the full form “Pharmacy Council of Bangladesh putting the abbreviation (PCB) within parenthesis before using the short form “PCB”. Please see line 70.
- Material and Methods in Abstract section require more data to give readers a better insight into this qualitative study (e.g. number of included respondents, study duration timeframe, etc).
Response: We highly appreciate this constructive feedback. We have revised the method section of the qualitative study in the abstract for a better clarity as below and inserted these revisions in lines 21-26.
“Methods: In-depth interviews (IDI) were conducted with graduate pharmacists (Grade A) and diploma pharmacists (Grade B) recruited from a few selected MPs that were included in a previous study. Key informant interview (KII) was conducted with the government and non-government stakeholders who were involved in pharmacy regulations and practices. Trained qualitative researchers conducted IDIs and KIIs using interview topic guides under relevant themes developed by the study investigators.
- The terms “IDI” (In Depth Interview) and “KII” (Key Informant Interview) must be described in more detail when they are mentioned for the first time in the Material and Methods section.
Response: Thank you for your feedback. We have described the IDI-In-Depth Interview and KII-Key Informant Interview in the Material and Method section. Please see the elaborated section on IDI as below in lines 110-112.
“We have purposefully selected from the list of eligible Grade A and Grade B pharmacists to have a good mix of age, sex, location, and duration of employment before inviting them to participate in an in-depth interview (IDI).”
To elaborate KII, we have elaborated the write up in lines 112-118 as below.
“Simultaneously, an additional list of the government and non-government stakeholders was created in consultation with the DGDA for conducting key informant inter-views (KII) including those stakeholders who have been involved in regulatory, administrative, practice, and human resource development for pharmacy practice in Bangladesh; such as, representatives of the DGDA, PCB, institutions offering degrees or diploma in pharmacy, and other relevant organizations.”
- In the Material and Methods section, it is necessary to describe the time period when the research was conducted.
Response: Thank you for your feedback. We have added the time period of the study as below. Please see lines 26, 166.
Line 26: “Between February and March 2021, nine Grade A and six Grade B pharmacists and nine government and non-government stakeholders were interviewed.”
Line 165: “Between February to March 2021, IDIs were conducted with nine Grade A pharmacists and six Grade B pharmacists and Table 1 summarizes their socio-demographic characteristics.”
- Line 137-138. This sentence requires correction to make it more understandable.
Response: Thanks for pointing this out. We have revised the sentence and made it clearer. Please see the revised sentence below (line 163-164).
“A written informed consent was obtained from each respondent prior to enrolment in the study and before conducting an interview.”
- The Ethics approval subsection in Materials and Methods is missing the numbers and dates under which decisions granting consent to conduct research were recorded.
Response: We highly appreciate pointing out this very important note on the methods. Now we have added the ethics approval information as below and inserted the revisions in line 160-163.
“Ethics approval was obtained from the Ethical Review Committee (ERC) of Inter-national Centre for Diarrhoeal Disease Research, Bangladesh (icddr,b) (PR-20142, 15 February 2021) and the National Research Ethics Committee of Bangladesh Medical Research Council (BMRC).”
- The first few sentences in the Discussion section should contain the purpose of the research and the author's summary of the study that was done.
Response: We have further revised the discussion section summarizing the purpose of the research as below. Please see lines 430-432.
“The purpose of the study was to explore the challenges and opportunities of good pharmacy practice in Model Pharmacies from the perspectives of the pharmacists and other relevant stakeholders.”
- Line 414-415. Sentence "Additional unsatisfactory aspects such as- low salaries, poor social status, lack of recognition, and long working hours." it seems that it is not complete and finished.
Response: We sincerely apologize for leaving this unintentional error. Now we have revised the sentence as below to improve clarity of the statement. Please see lines 432-435.
“Grade A pharmacists face several challenges that hinder their motivation and job satisfaction. These include juggling multiple responsibilities, receiving inadequate salaries, having a poor social status, working in unfavourable environments with long hours, and experiencing a lack of recognition and respect for their profession.”
- The Conclusions section should follow the research results, which means it should be more detailed.
Response: We highly appreciate this suggestion. Now we have revised the conclusion section and highlighted the research results as below. Please see lines 516-525.
“This study has highlighted the challenges faced by of Grade A and Grade B pharmacists working in model pharmacies. The model pharmacy initiative is a unique and promising effort to promote good pharmacy practice across Bangladesh. However, its success would depend on the rigorous monitoring, consistence enforcement and support from the relevant authorities to ensure adherence to established standards. Despite challenges the model pharmacy initiatives have great potential for further improvements by enhancing regulatory standards and effective implementation, and fostering a supportive work environment in MPs for retaining the pharmacists. Periodic training, better incentives, and the provision of heightening social value of the pharmacists would be essential for empowering the pharmacists for delivering a high-quality pharmacy services towards better health outcomes for the people in Bangladesh.”
Reviewer 2 Report
Comments and Suggestions for Authors
The title of the manuscript is consistent with the content of the manuscript.
The main question addressed by the researcher is characterization of the Model Pharmacies (MPs) system, which is newly established in Bangladesh, from different perspectives.
In the introduction section are presented different aspects regarding the profession of pharmacy in Bangladesh. In addition: a brief presentation of the local health system characteristics is necessary: about the regulations to allow pharmacists to prescribe independently; about the OTCs vs prescription drugs; pharmacy authorization; who can be stakeholders/owners in a pharmacy; etc.
Is the model pharmacy a pharmacy chain?
Rows 47-49: in addition: the competencies of pharmacist’ vary from country to country
Row 63: explication is necessary, for readers not familiarized with the local circumstances
Row 75-76: What is the difference between a model pharmacy and a regular pharmacy?
In the Materials and methods section are properly described the research procedure.
Row 115: the abbreviation KII must be explained at the first mention in the text.
Discussion:
Row 442: use of expression: training/education of pharmacists is recommended.
The presented Model Pharmacy system can be compared with other similar studies.
Strength and limitations
Among the limitations of the study should be emphasized the low number of the interviews. It is not mentioned if this number (a total of nine Grade A pharmacists and six Grade B pharmacists) what percentage is of the total number of pharmacists, working in the Model Pharmacies.
The References are appropriate.
Annex 2: the text is not readable in the column 2 – rows 2 and 3 cells
There are text-editing mistakes: space before the square-bracket; Sentence capital letter the first letter, etc.
Author Response
Response to Reviewer 2 Comments
We appreciate the comments and feedback made by reviewer 2. We have responded to each comment raised by the reviewer 1 and addressed the specific feedback point by point. Our responses are structured as follows:
a) Comments from the reviewer are presented in bold, and
b) Our responses are provided directly under each comment.
Comments and responses:
- The title of the manuscript is consistent with the content of the manuscript.
Response: We appreciate the reviewer comment.
- The main question addressed by the researcher is characterization of the Model Pharmacies (MPs) system, which is newly established in Bangladesh, from different perspectives.
Response: We highly appreciate understanding of the reviewer.
- In the introduction section are presented different aspects regarding the profession of pharmacy in Bangladesh. In addition: a brief presentation of the local health system characteristics is necessary: about the regulations to allow pharmacists to prescribe independently; about the OTCs vs prescription drugs; pharmacy authorization; who can be stakeholders/owners in a pharmacy; etc.
Response: We appreciate the feedback. We have added a paragraph (lines 75-83) describing the health system perspective of the pharmacy services, regulatory authority and OTC and prescription drugs and pharmacy owner issues. Please see the added paragraph below-
“The Directorate General of Drug Administration (DGDA) is the government regulatory body for ensuring quality pharmacy practice in Bangladesh. In Bangladesh, independent prescribing rights for pharmacists is limited except over-the-counter (OTC) drugs. There are 39 OTC drugs in Bangladesh [14]. As per the law of Bangladesh, any individuals having a valid pharmacist registration either Grade A, or Grade B or Grade C can apply for license to DGDA for opening a retail pharmacy and sell OTC and prescription medicines. These regular pharmacies often operate with less over-sight, lacking graduate pharmacists, sometimes sell prescription medicines without prescriptions, absence of counselling services, and often run by Grade C pharmacist [15-17]”
- Is the model pharmacy a pharmacy chain?
Response: Model pharmacy is a medicine shop that maintains standards as per the guideline outlined by the Directorate General of Drug Administration in Bangladesh. Now we have updated the paragraph on model pharmacy in the “Introduction” as below. Please see line 84-99.
“In 2015, DGDA has launched accreditation of two new levels of medicine outlets, Model Pharmacy and Model Medicine Shop, in order to promote good pharmacy practice (GPP). A Model Pharmacy is managed, served, and supervised by a graduate pharmacist (Grade A) with the support of a lower grade pharmacists, such as Grade B [18]. Between 2016 and 2020, DGDA inaugurated 274 model pharmacies in Bangladesh to ensure high-quality pharmacy services requiring presence of graduate pharmacists and adherence to the strict guidelines for dispensing medicines (e.g.; labelling), storage, pharmacy-grade refrigerator, air conditioning system, adequate space (at least 300 square feet), patient counselling service, etc. MPs are designated to promote safe pharmacy practices and prevent misuse of antibiotics [18, 19] whereas ordinary pharmacies are often operated with less oversight, lacking graduate pharmacists, absence of counselling services, and often run by the Grade C pharmacists [12, 15].”
- Rows 47-49: in addition: the competencies of pharmacist’ vary from country to country.
Response: Thank you very much for drawing our attention to this important aspect. Now we have revised the sentence to reflect the concept about variation ins competency of pharmacists as below. Please see the revised text below in lines 49-52.
“Pharmacists have to perform several roles, including filling prescriptions, handling orders, checking inventory, maintaining patient’s record, counselling patients, etc. and these roles vary from country to country based on the level of competencies [5, 6].
- Row 63: explication is necessary, for readers not familiarized with the local circumstances.
Response: We appreciate your feedback. To provide greater clarity on the types of pharmacists and regulatory authorities, we have included a narrative in the "Introduction" section (lines 84–99) and referenced it in our response to question 3. Please see the paragraph above under the response to question 3. We hope this will help in familiarizing you with the local circumstances of the pharmacists and their regulatory authorities.
- Row 75-76: What is the difference between a model pharmacy and a regular pharmacy?
Response: We appreciate the comment of the reviewer. We already have provided a narrative to clarify a similar query in our response to the comment 4 above.
- In the Materials and methods section are properly described the research procedure.
Response: We highly appreciate endorsement of the learned reviewer.
- Row 115: the abbreviation KII must be explained at the first mention in the text.
Response: We apologize for this unintentional omission. We have now added the full form before introducing the abbreviation "KII" (lines 112–118). Please see the elaborated section.
“Simultaneously, an additional list of the government and non-government stakeholders was created in consultation with the DGDA for conducting key informant inter-views (KII) including those stakeholders who have been involved in regulatory, administrative, practice, and human resource development for pharmacy practice in Bangladesh; such as, representatives of the DGDA, PCB, institutions offering degrees or diploma in pharmacy, and other relevant organizations.”
Discussion:
- Row 442: use of expression: training/education of pharmacists is recommended.
Response: We appreciate the reviewer's comment. To make a clear expression, we have now revised the sentence as below. Please see lines 475-477.
“Grade A pharmacists and stakeholders were opposed to the idea of placing Grade B pharmacists as in-charge in model pharmacies considering the insufficient training and inadequate curriculum of Grade B pharmacists.”
- The presented Model Pharmacy system can be compared with other similar studies.
Response: Thank you. We have added a sentence to compare the model pharmacy system with other similar studies. Below is the inserted sentence (lines 495-498).
“This type of accreditation of medicine outlets was found to be scalable and sustainable in Tanzania, as it had effectively improved the quality of pharmacy services, ensured better adherence to regulatory standards, enhanced customer trust, and contributed to improved healthcare outcomes [31, 32]”
Strength and limitations
- Among the limitations of the study should be emphasized the low number of the interviews. It is not mentioned if this number (a total of nine Grade A pharmacists and six Grade B pharmacists) what percentage is of the total number of pharmacists, working in the Model Pharmacies.
Response: Thank you for pointing this out. In qualitative studies, a proportional sample size is not required for representing the study population; rather, it is more important to achieve data saturation. Although we interviewed only six Grade B pharmacists, we achieved data saturation. We hope this adequately addresses the sample size concerns regarding Grade B pharmacists.
- The References are appropriate.
Response: We appreciate endorsement of the reviewer.
- Annex 2: the text is not readable in the column 2 – rows 2 and 3 cells
Response: We apologize for the technical error. We have now extended the table to improve its readability.
- There are text-editing mistakes: space before the square-bracket; Sentence capital letter the first letter, etc.
Response: We apologize for the editorial issues remained in the manuscript. We have now thoroughly edited the manuscript. We apologize for the editorial issues that remained in the manuscript. We have now thoroughly revised and edited it.
Reviewer 3 Report
Comments and Suggestions for Authors
The manuscript presents the results of a qualitative study designed to understand the challenges and opportunities of model pharmacy services in Bangladesh from both pharmacists` and stakeholders` perspectives. The manuscript is well written, with alignment between data collection, analysis and results of the study. The article could be accepted after some minor revisions:
- The abstract is a bit too long, the information presented here could be more concise.
- Abbreviations must be explained where they appear for the first time in the text (eg KII and IDI).
- Also, the strength and limitation section need to be expanded, for example another limitation of the study could be that the pharmacists who were interviewed represent only particular age categories, between 20-24 and 25-29 years, respectively.
Author Response
Response to Reviewer 3 Comments
We appreciate the comments and feedback made by reviewer 3. We have responded to each comment raised by the reviewer 1 and addressed the specific feedback point by point. Our responses are structured as follows:
a) Comments from the reviewer are presented in bold, and
b) Our responses are provided directly under each comment.
Comments and responses:
- The manuscript presents the results of a qualitative study designed to understand the challenges and opportunities of model pharmacy services in Bangladesh from both pharmacists` and stakeholders` perspectives. The manuscript is well written, with alignment between data collection, analysis and results of the study. The article could be accepted after some minor revisions:
Response: We appreciate the reviewer comment.
- The abstract is a bit too long, the information presented here could be more concise.
Response: We appreciate the reviewer feedback and comment. Now we have revised the “Abstract” and made it more concise (lines 17-40). Please see the revised abstract below.
“Background: In 2015, the Directorate General of Drug Administration (DGDA) of Bangladesh accredited Model Pharmacies (MPs) to enhance the quality of pharmacy services across the country. We examined the challenges and opportunities of pharmacists in MPs, and also explored the perspectives of the pharmacy stakeholders for improving good pharmacy practices (GPP) in Bangladesh. Methods: In-depth interviews (IDI) were conducted with graduate pharmacists (Grade A) and diploma pharmacists (Grade B) recruited from a few selected MPs that were included in a previous study. Key informant interview (KII) was conducted with the government and non-government stakeholders who were involved in pharmacy regulations and practices. Trained qualitative researchers conducted IDIs and KIIs using interview topic guides under relevant themes developed by the study investigators. Results: Between February and March 2021, nine Grade A and six Grade B pharmacists and nine government and non-government stakeholders were interviewed. The key challenges as well as demotivational factors for Grade A pharmacists were reported to be multiple responsibilities, inadequate salary, poor social status, unfavourable working environment, long working hours, lack of recognition and low respect for their profession. However, Grade B pharmacists have expressed job satisfaction, primarily due to working opportunities in reputable pharmacies and learning opportunities. The stakeholders reported a high operation cost of the MPs, shortage of trained pharmacists, poor salary structures and lack of public awareness about the critical roles of the pharmacists in healthcare to be challenges of retaining Grade A pharmacists at the MPs. Addressing the challenges of the pharmacists and revising compensation packages along with strengthening monitoring systems would be important for improving GPP at the MPs. Conclusion: The study has demonstrated that specifying the roles of the pharmacists, offering competitive packages, conducive working hours, and professional recognitions would be imperative for retention of trained pharmacists at MPs. Implementing regulatory standards and monitoring performance would enhance good pharmacy practice in Bangladesh.”
- Abbreviations must be explained where they appear for the first time in the text (eg KII and IDI).
Response: Thank you for your feedback. We have described the IDI-In-Depth Interview and KII-Key Informant Interview in the Material and Method section. Please see the elaborated section on IDI as below in lines 110-112.
“We have purposefully selected from the list of eligible Grade A and Grade B pharmacists to have a good mix of age, sex, location, and duration of employment before inviting them to participate in an in-depth interview (IDI).”
To elaborate KII, we have elaborated the write up in lines 112-118 as below.
“Simultaneously, an additional list of the government and non-government stakeholders was created in consultation with the DGDA for conducting key informant inter-views (KII) including those stakeholders who have been involved in regulatory, administrative, practice, and human resource development for pharmacy practice in Bangladesh; such as, representatives of the DGDA, PCB, institutions offering degrees or diploma in pharmacy, and other relevant organizations.”
- Also, the strength and limitation section need to be expanded, for example another limitation of the study could be that the pharmacists who were interviewed represent only particular age categories, between 20-24 and 25-29 years, respectively.
Response: Thank you for pointing this out. Pharmacists working at model pharmacies belong to specific age categories, as we observed that most of them are relatively young (under 30 years old). This is primarily because, the model pharmacy initiative is a relatively new program, established in 2016. As a result, the pharmacists recruited in this model pharmacies were predominantly in a particular age group. However, we tried to interview diverse characteristics of the pharmacists including age, sex, location and duration of employment which we described in the “Study design and settings” part (lines 110-112). We hope this adequately addresses the pharmacists particular age category concerns.